# The Potential and Limitations of Critical Raw Material Recycling: The Case of LED Lamps

**Julia S. Nikulski** [ID]**, Michael Ritthoff * and Nadja von Gries**

Wuppertal Institute for Climate, Environment and Energy, Doeppersberg 19, 42103 Wuppertal, Germany;
julia.nikulski@protonmail.com (J.S.N.); contact@nadjavongries.de (N.v.G.)
* Correspondence: michael.ritthoff@wupperinst.org

**Abstract:** Supply risks and environmental concerns drive the interest in critical raw material recycling in the European Union. Globally, waste electrical and electronic equipment (WEEE) is projected to increase by almost 40% until 2030. This waste stream can be a source of secondary raw materials. The determination of the economic feasibility of recycling and recovering specific materials is a data-intensive, time-consuming, and case-specific task. This study introduced a two-part evaluation scheme consisting of upper continental crust concentrations and raw material prices as a simple tool to indicate the potential and limitations of critical raw material recycling. It was applied to the case of light-emitting diodes (LED) lamps in the EU. A material flow analysis was conducted, and the projected waste amounts were analyzed using the new scheme. Indium, gallium, and the rare earth elements appeared in low concentrations and low absolute masses and showed only a small revenue potential. Precious metals represented the largest revenue share. Future research should confirm the validity and usefulness of the evaluation scheme.

**Keywords:** recycling economics; urban mining; LED lamps; material flow analysis



## 1. Introduction

The annual global extraction of primary resources has grown almost fourfold between 1970 and 2010 and is significantly contributing to the loss of biodiversity, water stress, and climate change [1]. Wiedenhofer et al. [2] found that 53% of the global materials processed in 2014 entered the anthropogenic stock as part of buildings, technical infrastructure, durable consumer goods, or other long-lasting products. This ratio increased from 16% in 1900. Forecasts show that even considering efforts to stabilize the use of global primary stock-building materials total in-use material stocks will more than double between 2014 and 2050 worldwide [2]. These high raw material inputs are caused by current production and consumption patterns in linear economies [3]. The EU economy's growth is increasingly dependent on non-energy raw materials—such as metals and minerals—whose criticality was previously paid less attention to than that of oil and gas [4]. The European Commission estimated that roughly 30 million European jobs are contingent on the availability of raw materials [5]. According to a 2014 report by the European Commission [4] on critical raw materials, not only is the economic importance of some of these materials high, but also their supply risk. Around 91% of the overall non-energy raw materials used in the EU28 are imported from outside the member states. This means that procurement dependencies from countries with unstable governance systems (e.g., a weak rule of law, high levels of corruption, and political instability) can increase the uncertainty of material availability and jeopardize growth and jobs in Europe [4]. This supply risk could be lowered if critical materials were substituted or if materials were recycled from End-of-Life (EoL) products [4,6]. These economic as well as ecological concerns therefore lead to a growing interest in the resource potential of anthropogenic stockpiles and the recovery of secondary raw materials.

Within these stockpiles, electrical and electronic equipment (EEE)—such as monitors, lamps, and large and small household appliances—is a fast-growing portion. When this equipment reaches the end of its lifetime, it moves from stock to waste [7]. Forti et al. [7] estimated that in 2019 roughly 53.6 Mt of e-waste was generated globally. This amount is projected to increase to 74.7 Mt in 2030. Yet, only a fraction of this waste is collected and recycled, leaving valuable materials unrecovered in municipal solid waste or landfills [7]. The 2012 EU directive on waste electrical and electronic equipment (WEEE) addressed this issue by setting minimum standards and ratios for the collection, recycling, and recovery rates of e-waste [8]. Collection systems need to be expanded, and recycling technologies need to be enhanced or newly developed to achieve these targets.

Whether material recycling from the anthropogenic stock is ecologically and economically feasible compared with the extraction from primary raw materials is usually answered based on specific case studies. For this purpose, the case-specific processes for the extraction of the primary and secondary materials are then evaluated and compared with each other. Such a procedure, however, is time-consuming and linked to one specific technology. In many cases, no processes have yet been established for the recycling of materials and the production of secondary raw materials.

Light-emitting diodes (LED) lamps are an example of EEE with a relatively long lifespan and growing production, consumption, and EoL flows [9]. Yet, there is currently no established recycling technology available for LED lamps [10]. Only a few studies have investigated the recycling and material recovery of LED lighting. While some focused on technological development [11–13], others assessed the environmental impacts of LED lamps and their EoL phase [14] or discussed the economic potential of material recovery [9,15,16]. Studies analyzing the economic viability of LED lighting and other WEEE streams often focused on the absolute raw material amounts in the EoL products combined with their prices to derive recommendations for actions [16–18], or they focused on a cost–benefit analysis [19,20]. However, cost–benefit calculations require a significant amount of data input and often refer to specific recycling technologies. Meanwhile, a sole focus on raw material prices allows selecting the most profitable materials out of the ones considered. Still, it neglects the larger context and question of whether the amounts contained in the waste stream warrant to be recycled given their concentration in the overall amount of waste.

The question can be raised whether an initial assessment of recycling feasibility is possible based on simple and generally accessible information. Therefore, this paper introduces a two-part evaluation scheme to conduct such an initial assessment of economic feasibility for the material recycling of any WEEE stream. This scheme can be considered a precursor to cost–benefit analyses. It allows assessing the viability of recycling independent of a specific technology by evaluating two areas: First, material concentrations in the total LED lamp waste were compared to average material concentrations in the earth's crust, specifically the upper continental crust concentrations reported by Rudnick and Gao [21]. Second, raw material prices combined with the total amounts of materials embedded in the waste streams indicated which materials would generate the highest potential revenues. The main objective of introducing this evaluation scheme is to provide a method that can estimate the economic feasibility of recycling in a relatively fast and easy way by leveraging only easily accessible data. The results from this evaluation could indicate whether further investigations into new recycling methods are warranted and on which materials to focus.

LED lamp recycling was investigated in the European Union to illustrate the application of this evaluation scheme. This paper is structured as follows: Section 2 provides an overview of the methodology used to forecast the LED lamp waste generation between 2017 and 2030 in the EU28 member states. The data and the Weibull distribution used to model the future waste streams are described, and the proposed evaluation scheme is presented. Section 3 shows the results of the LED lamp waste forecast and the amounts of materials embedded in this waste stream. Using these results, the total potential revenue per material is calculated. The material concentration of the entire LED lamp waste is

compared with the upper crust concentrations for each raw material. Section 4 discusses the results to determine whether the introduction of a new recycling technology would be feasible. Section 5 summarizes the findings of this study.

## 2. Materials and Methods

### 2.1. Data Collection

The following data were gathered to model the projected waste flows of LED lamps for the EU28 member states between 2017 and 2030. The material composition of white LEDs per average die area and the LED die area per LED lamp were used to calculate the mass of the specific material per lamp. The put on the market data, the average lifespan, and the average weight of LED lamps were combined to calculate the generated LED lamp waste. Applying the specific material per LED lamp to the total LED lamp waste yielded the total material weight contained in the waste. The consideration of collection, recycling, and material recovery rates for LED lamps allowed determining the recycling feasibility based on system and thermodynamic restrictions limiting the amount available for recovery.Because supply risk is a significant factor motivating the recovery of secondary raw materials, this case study focused on the materials included in LEDs, which are categorized as "critical" by the EU: cerium, europium, gadolinium, gallium, indium, palladium, terbium, yttrium [6]. Gold and silver were also included, given their high total material requirements (TMR) [22]. The relevant parts of an LED lamp that contain these materials are the chip, the interconnection technology, the phosphorus, and the printed circuit board (PCB) of white LEDs [15,23]. Different studies investigated the material composition of LED lamps with considerable differences in the reported amounts (e.g., [15,23–26]). The bill of materials used for this analysis was primarily taken from Deubzer et al. [23] and Buchert et al. [15]. They provided the most comprehensive list of critical raw materials included in LEDs. The weights for the materials included in the PCB were derived using material ratios published by Huisman et al. [27]. Those were applied to the weight of a PCB per one unit of LED reported by Scholand and Dillon [28]. The weights of the rare earth elements related to different types of phosphorus: YAG:Ce, TAG:Ce, ortho-silicate, or GAG:Ce [15,23]. The exact share of white LED lamps per phosphorus type was unknown. Therefore, the calculations in this study considered all rare earth elements that could potentially—with the given concentrations—be contained in a white LED. This was taken into account during the interpretation of the results. Bond wiring combined with gluing was assumed to be the most common interconnection technology and included in the bill of materials [23]. An overview of all materials considered is shown in Table 1.

**Table 1.** Material demand for selected LED lamp components and critical raw materials relating to 1 mm$^2$ die area of white LED.

| Component | Material | Weight (mg) |
|:---:|:---:|:---:|
| Chip | Gallium | 0.007 [23] |
| Chip | Indium | 0.009 [23] |
| Phosphorus | Cerium | 0.003 [23] |
| Phosphorus | Europium | 0.003 [23] |
| Phosphorus | Gadolinium | 0.015 [15] |
| Phosphorus | Terbium | 0.165 [23] |
| Phosphorus | Yttrium | 0.089 [23] |
| Printed circuit board | Gold | 0.155 [27,28] |
| Printed circuit board | Silver | 1.703 [27,28] |
| Printed circuit board | Palladium | 0.093 [27,28] |
| Interconnection technology | Gold | 0.019 [23] |
| Interconnection technology | Silver | 0.276 [23] |

The number of LED lamps put on the market (POM) in EU28 member states until 2030 was calculated using data from Marwede et al. [9], who estimated the development of POM amounts between 2008 and 2020. Buchert et al. [26] approximated that white LED

lamps would have a market share of 95% in 2025 and be partially displaced by white OLED lamps until they would reach 75% market share in 2050. Based on this, we assumed that the growth of market share would slow until 100% is reached in 2030, and afterward decline. Figure 1 shows the overall EU-trend of POM amounts for white LED lamps. Table A1 shows the POM data used to derive this figure.

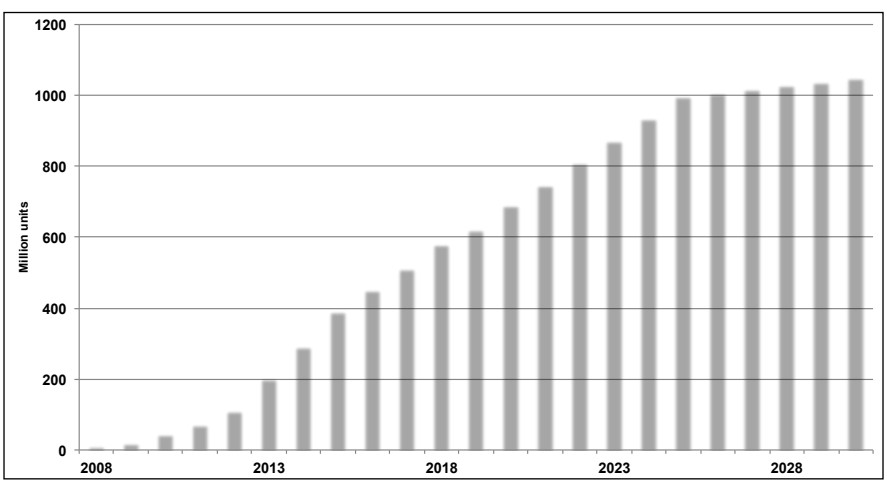

**Figure 1.** Development of put on the market amounts of LED lamps in EU28 countries until 2030.

The average lifespan, the average weight, and the average LED die area depend on different application types of lamps. Besides the application of LEDs in general lighting, there are many other applications of LEDs, such as in backlights of displays of electrical appliances. However, in this study, we focused on the application of LEDs in general lighting, which dominates the LED market [23]. Marwede et al. [9] differentiated between residential lamps and retrofits, commercial lamps and retrofits, industrial, outdoor, and architectural lamps based on common market segments. Data on average lifespans, weight, and die area for LED lamps is sparse. The length of the domestic service lifespan—describing the time from shipment of a product to its first user until the time of discard by its last user [29]—is difficult to determine for LED lamps. Therefore, technical lifespans combined with typical operating hours reported by Marwede et al. [9] were used. The average weight of LED lamps differed by application and was based on individual case studies. Aside from the case of residential retrofits [30], no studies were found which investigated the weight of several different lamps of the same application type. The average size of the LED die area per LED lamp determines the total amounts of critical raw materials included in one lamp. This size varies depending on different applications, and very few studies previously dealt with the determination of this size [15,23]. Deubzer et al. [23] used differentiated die areas for each application type with a lower and upper limit. For this study, the weighted average of these die areas was used. Table 2 shows the average lifespan, weight, and die area for each application type used in this study.

Collection rates of LED lamps per EU28 member state were approximated using Eurostat data on the collection rate of lighting equipment because there are no data available on the collection amounts of LED lamps. These data are shown in Table A2. The latest available data for the EU28 were used to extrapolate the development of the rate until 2030. Scenario 1 assumed that the 2017 collection rate across all EU28 member states would remain the same until 2030 at 14%. Scenario 3 assumed that the EU target of a collection rate of 85% would be achieved [7], while scenario 2 represented achieving half of the 85% target by 2030. The EU collection rate target of 85% relates to the amount collected compared to the total waste generated in a member state in a given year [7]. No recycling rate for LED lamps is currently captured because there does not exist an LED recycling technology. Therefore, the recycling rate baseline scenario (scenario 1) was linked to the application of current recycling technologies to the recycling of LED lamps.

Table 2. The average lifespan, average weight, and average die area for different LED lamp types.

| Application Type | Average Lifespan (Years) | Average Weight (kg) | Average Die Area (mm$^2$) |
|---|---|---|---|
| Residential lamps | 18.8 [9] | 0.520 [31] | 9 ± 2 [23] |
| Residential retrofits | 12.5 [9] | 0.2452 [30] | 9 ± 2 [2] |
| Commercial lamps | 5.9 [9] | 1.75 [14] | 11 ± 3 [23] |
| Commercial retrofits | 2.9 [9] | 0.2452 [1] | 11 ± 3 [2] |
| Outdoor | 10.0 [9] | 15.0 [32] | 17 ± 4 [23] |
| Industrial | 8.3 [9] | 3.5 [23] | 40 ± 9 [23] |
| Architectural | 10.0 [9] | 4.5 [23] | 79 ± 42 [23] |

[1] Assumption that commercial retrofits weigh the same as residential retrofits. [2] Assumption that residential and commercial retrofits have the same average die area as lamps.

Reuter and van Schaik [33] simulated how much of the different materials included in LED lamps could be recycled in pre-processing steps depending on the LED lamp design to achieve a metal-rich fraction. While they did not report an overall recycling rate for the entire lamp, they did publish material-specific rates. For aluminum, for example, they predicted a recycling rate of around 75%, while the rate for copper was between 40% and 45%. These values were derived from an optimized recycling process aimed at increasing the recovery of the metal fraction contained in LED lamps [33]. Thus, the assumed baseline scenario with an overall recycling rate of 50% was lower than the reported values by Reuter and van Schaik to account for a non-specific technology applied. This rate assumed that 50% of the weight of the LED lamp could be recycled using current technologies. Scenario 3 reflected the achievement of the 80% target for recycling of WEEE [7]. In scenario 2, the recycling rate reached 65%, assuming that the newly developed technology could only reach the target halfway. The yearly recycling and collection rates between 2017 and 2030 for scenarios 2 and 3 resulted from interpolating the values in scenario 1 as starting values and the scenario values of 2 and 3 as end values. This approach represents continuous progress in expanding collection systems and optimizing recycling technologies. The different scenarios are shown in Table 3. The interpolated collection and recycling rates for each year are shown in Table A3.

Table 3. Collection and recycling scenarios of LED lamp waste for all EU28 member states in 2030.

| | Scenario 1 | Scenario 2 | Scenario 3 |
|---|---|---|---|
| Collection rate (%) | 14 | 50 | 85 |
| Recycling rate (%) | 50 | 65 | 80 |

After the collection and recycling steps, the retained materials need to be recovered. How much of each material is recovered depends on whether the metallurgical processes applied are compatible with each other, as well as the chemistry and the concentration of the materials [33]. As shown in [33], the Metal Wheel illustrates the different recovery paths for materials included in EoL products. The recycling steps used to separate elements and components of products into different fractions determine which metallurgical processes would be applied and which materials would be lost or recovered. The different metal routes shown in the Metal Wheel illustrate the incompatibility of the recovery of gallium and indium with rare earth elements, as well as the limited possibilities to recover precious metals in the same process as either rare earth elements or indium/gallium [33]. Information on specific material recovery rates for LED lamps is rare. However, the limited available research was used to derive different groups for the recovery of different critical raw materials from LED lamp waste. Each group focused on a different combination of materials that were compatible with each other and could be extracted using a particular process. The compatibility of the materials and the potential recovery rates were derived from studies investigating the recovery of these materials from LED lamps or compact fluorescent lamps (CFLs). Table 4 provides an overview of the recovery rate groups.

**Table 4.** Material recovery groups are derived for different critical raw materials based on values reported in cited studies.

| | Group A—Indium and Gallium Recovery | Group B—Rare Earth Elements Recovery | Group C—Precious Metals Recovery |
|---|---|---|---|
| Gallium | 90–99% [11–13] | – | – |
| Indium | 95% [11,34] | – | – |
| Cerium | – | 60–100% [35–37] | – |
| Europium | – | 90–100% [35–37] | – |
| Gadolinium | – | 50% [37] | – |
| Terbium | – | 77% [37] | – |
| Yttrium | – | 76–100% [35–37] | – |
| Gold | – | 38–68% [36] | 50–100% [33,38] |
| Silver | – | – | 50–81% [33,38] |
| Palladium | – | – | 13–60% [33,38] |

### 2.2. Data Modeling

According to Oguchi et al., the amount of LED waste generated $G_t$ at the end of year $t$ could be calculated by the product sum of the amounts of products put on the market (POM) between years $i$ and $t - i$, multiplied by the average weight $w$ of an LED lamp and the percentage of LED lamps discarded in year $t$, expressed by $f_t(i)$ [39]. It was assumed that the average weight of an LED lamp is constant over time. The corresponding equation proposed by Oguchi et al. [39] is

$$G_t = \sum_i POM_{t-1} \cdot w \cdot f_t(i), \tag{1}$$

$$\text{where } f_t(i) = W_t(i + 0.5) - W_t(i - 0.5), \tag{2}$$

$$\text{and } W_t(y) = 1 - \exp\left[-\left\{\frac{y}{y_{av}}\right\}^k \cdot \left\{\Gamma\left(1 + \frac{1}{k}\right)\right\}^k\right]. \tag{3}$$

To calculate the share of lamps leaving the use phase, the lifetime distribution of LED lamps had to be modeled, which was carried out with the help of the cumulative Weibull distribution function $W_t$ [40]. This function, given by Equation (3), is defined by the average lifespan of LED lamps $y_{av}$, the shape factor $k$ of the distribution, the lifespan $y$, and the gamma function $\Gamma$ [39]. The Weibull function is a common distribution function to model data on product survival [40]. Factor $k$ determines when the majority of the LED lamps are discarded: a small value indicates early disposal, while a large value signifies that products remain longer in use [41]. No previous studies were found which reported $k$ values for LED lamps. Therefore, studies investigating other EEE were reviewed to choose an approximate value for $k$. Kalmykova et al. [42] collected lifespan data from discarded LED TVs and monitors. The reported $k$ for LED TVs of 3.75014 was used in this study, implying that LED lamps are more likely to be used until the end of their technical lifespan compared to being disposed of early due to consumer preferences. The materials included in the calculated LED waste $G_t$ were determined based on the units of LED lamps discarded and the materials contained in one unit of LED lamp.

### 2.3. Data Evaluation

The economic feasibility criteria introduced in this study are comprised of the upper crust material concentrations and the raw material prices. The natural accumulation of usable minerals and rocks is called a deposit if the exploitation of this accumulation can be realized economically depending on its size and contents [43]. There is no comparable definition for the materials contained in the anthropogenic stock. Therefore, all amounts of all materials in this stock are usually considered to be anthropogenic deposits (e.g., [44]). Yet, similar to natural deposits, anthropogenic deposits need to be judged based on their size and contents to determine whether the contained resources are mineable and exploitation is economically feasible. Moreover, this assessment of anthropogenic deposits needs

to be compared to natural deposits to decide if secondary raw materials are ecologically and economically favorable over primary resources. In this context, it can be worthwhile to draw on information introduced in geosciences. Various researchers investigated and determined the concentrations of elements in the upper continental crust. The best-known results came from Clarke and Washington [45] as well as Goldschmidt [46]. However, these publications do not cover all elements, and specifically, they lack values for the trace elements important to our study. Further investigations and calculations followed, and data on trace elements were gathered. In this study, reference is therefore made to the relatively recent work of Rudnick and Gao [21].

In order to form an orebody, the element under consideration must be enriched many times above the normal abundance in the earth's crust. The minimum content of a mineable deposit and the degree of enrichment—called enrichment factor—differ between various elements [47]. The limit of the economic feasibility for mining would only be below the average content of the earth's crust under exceptional circumstances, e.g., when particularly simple and cost-effective processing is possible. This applies, for example, to the extraction of titanium raw materials from marine soaps [43], which is a rare exception. Thus, the concentration of elements in the upper crust of the earth can usually be considered as the lowest limit for the extraction of raw materials. Furthermore, the various elements in the anthroposphere are usually not easily accessible and separable, but rather, they are often present in complex products and material compounds. Therefore, it can be assumed that the material recovery from the anthropogenic stockpile is significantly less ecologically and economically advantageous compared to the extraction from enriched natural deposits with higher concentrations.

In addition to the upper crust concentrations, raw material prices were used for the evaluation to determine the overall revenue potential. The data availability for raw material prices is scattered. Two different sources relating to slightly different time horizons needed to be considered to calculate revenue estimations for all critical raw materials included in this study. Data from the United States Geological Survey (USGS) for the year 2019 was used for gold, silver, palladium, indium, gallium, and yttrium [48]. For cerium, europium, gadolinium, and terbium, data from the Institute for Rare Earths and Strategic Metals (ISE) were collected to calculate yearly averages between October 2019 and September 2020 [49–60]. An overview of the considered prices is given in Table A4.

### 3. Results

*3.1. Total Mass of Critical Raw Materials Included in LED Lamp Waste between 2017 and 2030*

According to our calculations, more than 2.6 million tons of LED lamp waste would be generated between 2017 and 2030 in the EU28 member states. That corresponds to 2.4 million LED lamps that would be discarded. Considering the average weight and material composition per lamp, Table 5 shows the total mass of critical raw materials included in the generated waste. Silver was the material with the highest mass contained in this waste with more than 60 t. At the same time, cerium and europium had the lowest shares with only 91.02 kg. Precious metals accounted for the largest share of critical raw material mass. Indium and gallium, on the other hand, were contained in lower masses than rare earth elements. These amounts represented the theoretical potential for the recycling and recovery of critical raw materials included in the LED lamp waste. However, collection system limitations, recycling inefficiencies, and recovery process constraints reduce the actual amount that can be extracted from the waste. Therefore, how much of the total mass of materials can be recovered and used as secondary raw materials depends on the change of the collection rate, recycling rate, and the assumed material recovery rate.

**Table 5.** The theoretical potential of materials available for recovery in LED lamp waste generated between 2017 and 2030 in EU28 member states.

| Element | Total Mass in LED Lamp Waste (kg) |
|---|---|
| Gallium | 212.38 |
| Indium | 273.05 |
| Cerium | 91.02 |
| Europium | 91.02 |
| Gadolinium | 455.09 |
| Terbium | 5006.00 |
| Yttrium | 2700.21 |
| Gold | 5274.15 |
| Silver | 60,048.40 |
| Palladium | 2818.62 |

*3.2. Material Recovery Potential after Collection and Recycling Steps between 2017 and 2030*

Table 6 gives an overview of the potential for the material recovery from the waste amounts depending on the different collection and recycling rate scenarios. For example, the total amount of cerium included in the LED lamp waste between 2017 and 2030 was 91.02 kg. The maximum amount that could be extracted from the lamp waste only ranged between 6.42 kg and 43.41 kg, depending on the collection and recycling rate scenario. Recovering cerium would also mean that materials such as indium, gallium, palladium, and silver would be lost during the recovery process. The amount of cerium was small compared to how much silver could be extracted from the lamp waste. Of the more than 60 t of silver stored in this waste, only a maximum between 3.4 t and 23 t could be recovered. To determine which critical raw materials should be treated as a priority in the material recovery process and whether introducing a new recycling technology would be economically feasible, the upper continental crust concentrations and raw material prices were introduced as scenario evaluation metrics.

**Table 6.** Weights of materials in LED lamp waste that could be recovered between 2017 and 2030 in EU28 member states [1].

| | | Scenario 1 (kg) | Scenario 2 (kg) | Scenario 3 (kg) |
|---|---|---|---|---|
| Group A | Gallium | 13.49–14.84 | 46.68–51.34 | 90.59–99.65 |
| | Indium | 18.31 | 63.35 | 122.95 |
| Group B | Cerium | 3.85–6.42 | 13.34–22.23 | 25.88–43.14 |
| | Europium | 5.78–6.42 | 20.00–22.23 | 38.83–43.14 |
| | Gadolinium | 16.06 | 55.57 | 107.85 |
| | Terbium | 272.04 | 941.32 | 1826.95 |
| | Yttrium | 144.83–190.57 | 501.15–659.40 | 972.65–1279.80 |
| | Gold | 141.45–253.11 | 489.43–875.82 | 949.91–1699.83 |
| Group C | Gold | 186.11–372.22 | 643.99–1287.97 | 1249.88–2499.76 |
| | Silver | 2118.97–3432.73 | 7332.06–11,877.93 | 14,230.38–23,053.21 |
| | Palladium | 25.86–119.36 | 89.48–412.99 | 173.67–801.55 |

[1] Scenario 1: a 14 % collection rate and a 50% recycling rate for all years. Scenario 2: increasing collection rate from 14% to 50% and increasing recycling rate from 50% to 65% between 2017 and 2030. Scenario 3: increasing collection rate from 14% to 85% and increasing recycling rate from 50% to 80% between 2017 and 2030. Exact values for each year can be found in Table A3 in Appendix A.

*3.3. Evaluation of Economic Feasibility*

The previously estimated amounts of critical raw materials contained in the total LED lamp waste were compared to the material concentrations in the upper continental crust. Table 7 displays the ratio of these concentrations. They indicated which raw materials occur in higher concentrations in the waste than in the upper continental crust. Indium and terbium appeared only slightly more frequently in the LED lamp waste. Silver, gold, and palladium, on the other hand, were significantly more highly concentrated in the

waste. Cerium showed the lowest concentration in the lamp waste. In addition, gallium, europium, gadolinium, and yttrium all appeared less frequently compared to the upper continental crust. This implies that LED lamp waste is not an adequate urban mine for these types of materials. It would likely be more cumbersome and expensive to extract these low concentrations from the waste than from natural deposits or other waste products containing higher critical raw material volumes. For the materials with only slightly higher concentrations—indium and terbium have a ratio between one and three—it is questionable whether it would be lucrative to extract them. As previously mentioned, not all of these materials could be recovered in the same metallurgical processes. The ratios of concentrations can put into perspective how difficult it will be to extract certain materials from the waste, given that lower concentrations can increase the likelihood that materials will be lost during recycling processes [33]. However, to determine the economic feasibility of the recovery of certain elements, it is necessary to consider the total mass contained in the waste in combination with raw material prices.

**Table 7.** Upper crust concentrations, as reported by Rudnick and Gao [21], compared to critical raw material concentrations in the LED lamp waste in EU28 member states.

| Raw Materials | Upper Crust Concentration (ppm) | Concentration in LED Lamp Waste (ppm) | Ratio of Waste to Upper Crust Concentration |
|---|---|---|---|
| Gallium | 17.5 | 0.081 | 0.005 |
| Indium | 0.056 | 0.104 | 1.856 |
| Cerium | 63.0 | 0.035 | 0.001 |
| Europium | 1.0 | 0.035 | 0.035 |
| Gadolinium | 4.0 | 0.173 | 0.043 |
| Terbium | 0.7 | 1.906 | 2.723 |
| Yttrium | 21.0 | 1.028 | 0.049 |
| Gold | 0.0015 | 2.008 | 1338.701 |
| Silver | 0.053 | 22.863 | 431.368 |
| Palladium | 0.00052 | 1.073 | 2063.743 |

The calculated amounts of critical raw materials were used to determine the potential revenues that could be generated from the recycling of LED lamp waste. Scenario 2 was chosen as the most likely future development, considering that thirteen years to achieve the EU targets in 2030—as assumed in scenario 3—is little time. Table 8 shows the estimated revenue that could be generated if the LED lamp waste between 2017 and 2030 were collected and recycled according to scenario 2. Recovery group A—consisting of indium and gallium—yielded the lowest revenue with a maximum of USD 53,972. Group B generated between USD 22.8 MM and USD 40.2 MM. The largest proportion of this revenue came from gold (USD 22 MM–USD 39.4 MM). As previously mentioned, the masses of the different rare earth elements would not occur at the same time in the total LED lamp waste because they depend on the specific phosphorus used in the white LEDs. The recovery of terbium yielded USD 699,361. The remaining rare earth elements—cerium, europium, gadolinium, and yttrium—only amounted to between USD 24,380 and USD 30,438 in total. This was due to a combination of low masses and material prices. Cerium, gadolinium, and yttrium had the lowest prices per kg—less than USD 35. Cerium, europium, and gadolinium also had very low masses—all below 56 kg. Therefore, a lower share of TAG:Ce phosphorus would lower the revenue more significantly compared to a lower share of YAG:Ce. Irrespective of the different phosphorus applied, the rare earth elements contributed a maximum of 1.8% to the overall potential revenue of group B. The precious metals in group C provided the highest revenue with a range between USD 37.1 MM and USD 84.1 MM. Due to its lower price per kg compared to gold and palladium, the recovery of silver only accounted for 7% of the total group C revenue, even though silver had the highest share of mass.

**Table 8.** Estimated revenue that could be generated from LED lamp waste recycling in the EU28 between 2017 and 2030.

| Recovery Group | Raw Materials | Scenario 2 (kg) | Raw Material Price [1] (USD/kg) | Estimated Revenue (MM USD) |
|---|---|---|---|---|
| Group A | Gallium | 46.68–51.34 | 570.00 | 0.051–0.054 |
| | Indium | 63.35 | 390.00 | |
| Group B | Cerium | 13.34–22.23 | 4.58 | 22.8–40.2 |
| | Europium | 20.00–22.23 | 286.33 | |
| | Gadolinium | 55.57 | 27.94 | |
| | Terbium | 941.32 | 742.96 | |
| | Yttrium | 501.15–659.40 | 34.00 | |
| | Gold | 489.43–875.82 | 45,010.98 | |
| Group C | Gold | 643.99–1287.97 | 45,010.98 | 37.1–84.1 |
| | Silver | 7332.06–11,877.93 | 520.84 | |
| | Palladium | 89.48–412.99 | 48,226.05 | |

[1] Raw material prices for gallium, indium, yttrium, gold, silver, and palladium refer to the average annual prices in 2019. Prices for cerium, europium, gadolinium, and terbium are averages of the monthly prices between October 2019 and September 2020. Values are also shown in Table A4 in the Appendix A.

Considering the potential revenue and ratios of concentration and comparing these among the different materials, all rare earth elements aside from terbium generated negligible amounts of revenue while appearing in lower concentrations in the waste than in the upper continental crust. Although indium appeared in the LED lamp waste almost twice as frequently compared to the concentration in the upper crust, its potential revenue contribution of USD 24.705 was very small. Terbium had the second-highest ratio of concentrations. However, with a maximum revenue of less than USD 700,000, terbium appeared to be less significant than any of the precious metals. Even silver, which had a lower price per kg than terbium, could earn between USD 3.8 MM and USD 6.2 MM because its mass was between 7 to 12 times higher than that of terbium. Precious metals generated the highest revenue, appeared in significantly higher concentrations in the waste compared to the upper continental crust, and had some of the highest total masses in the LED lamp waste of all considered critical raw materials. Therefore, the focus of recycling and recovery should lie on precious metals if a new recycling technology is developed.

## 4. Discussion

The differences in recovered material mass across the scenarios were significant. They showed the effect of losses that could occur due to inefficiencies in collection systems and recycling processes. The results also illustrated that the recovery of rare earth elements yielded negligibly small masses and revenue potentials. Several other studies reached similar conclusions on the economic viability of rare earth elements, indium, and gallium, and the contribution of precious metals to the overall revenue potential. Cenci et al. [16] found that gold was the most important material to recover in terms of economic value. Cucchilla et al. [17] investigated WEEE other than LED lamps—including LCD and LED monitors, smartphones, and notebooks—and discovered that gold contributed to more than half of the potential revenue that could be generated from all of these products. Reuter and van Schaik's [33] recycling simulation of LED lamps disregarded indium, gallium, and rare earth elements entirely, focusing instead on metal-rich fractions. In general, gallium is difficult to recycle and recover because it appears in material compounds that are challenging to untangle, and the amounts it appears in are very small [61]. Ylä-Mella and Pongrácz [62] mentioned in connection with indium that low material concentrations in products and the loss of quality during the recycling process pose economic barriers to recycling. Similar issues surrounding the recyclability of rare earth elements were discussed by Balaram [63], who highlighted their occurrence in low amounts and the difficulty of separating the rare earth elements individually. Moreover, the cost of recycling these elements from any EoL products exceeds the potential revenue that could be generated from them and is therefore not economically feasible [61–63].

As previously mentioned, one of the EU's objectives is to reduce the supply risks of critical raw materials contained in WEEE. Even if the potential revenue from rare earth elements, indium, and gallium is small, their mass could still be relevant to decrease import dependencies from other countries. However, comparing the amounts of some of these critical raw materials contained in the total mass with the yearly consumption of these elements in Europe shows that they contribute insignificantly to reducing supply risk. For example, Germany's total annual gallium demand was estimated at 30–40 t in 2015 [61]. The gallium in the cumulated LED lamp waste between 2017 and 2030 would only account for around 0.1% of Germany's yearly consumption. Little information is available on the demand for indium and rare earth elements. Global indium production was estimated to be 790 t in 2013 [64], demonstrating the small impact the recovered indium from LED lamps would have. According to a communication from the European Commission [6], the EU has a 0% import reliance on indium. However, all of the rare earth elements considered in this study pose a 100% import reliance for the EU [6]. No final determination can be made on the relation between the availability of recovered rare earth elements to their annual demand in the EU. Considering the absolute mass of these elements in the LED lamp waste, only yttrium and terbium seem to be of a relevant size to affect the supply risk.

The environmental perspective is another reason why LED lamps should be recycled. According to the review of LCA studies on LED lamps conducted by Franz and Wenzel [65], the disposal phase accounts for up to 27% of the total environmental impact of an LED lamp. However, recycling or energy recovery of the lamp can also create an environmental benefit. Most LCA studies show the highest environmental burden during the use phase [65]. In some cases, recycling can be very energy-intensive and more environmentally harmful than natural resource mining because of complex recycling processes required to untangle material compounds in complex products, as suggested in the case of indium recycling from LCD screens [66,67].

Whether the main incentive for recycling is economic or environmental, during the product design process, producers should already consider the EoL phase to make disassembly of LED lamps as easy as possible and to enable the recovery of as much material mass as possible [30,33]. Such eco-design strategies can increase the efficiency of the recycling process, as well as the environmental benefit [16]. While Dzombak et al. [30] saw some improvements in LED lamp design over a period of seven years, for example, a lower overall material mass, the majority of the examined lamps were still not easy to disassemble and contained elements and materials hindering high levels of material recovery. Another option to reduce the environmental impact of products is to focus on material efficiency and use less new material through light-weight design [68].

The results of this study have several limitations. First, the data used for the calculations were based on scarce information available in the literature. Various assumptions had to be made about the average weight and lifespans of different LED lamp applications and the development of LED lamp sales numbers in the EU. The effect of different recycling and collection rates on the results was considered by applying different scenarios. Second, the limited availability of studies investigating the material recovery of LED lamps led to the derivation of exemplary recovery groups that cannot fully represent all possible material combinations. For example, it is likely that gold could be recovered in the same process with indium and gallium, as suggested by the Metal Wheel [33]. Third, this study only considers critical raw materials and disregards the revenue potential of non-critical materials such as aluminum, copper, tin, and plastics. Integrating these into the analysis might change what courses of action are derived and which material groups are most profitable. Fourth, the raw material prices used are the average prices for primary resources. It is debatable whether the same prices can be achieved for secondary raw materials. However, because it is more difficult to gain information on secondary raw material prices, the primary resource prices were used as an approximation. Finally, this study only considered the LED package in the lamp, but not the lamp housing, PCB, or other electronic parts,

disregarding further non-critical materials that could be recovered to be used as secondary materials and generate profits.

This study highlights several areas for future research. First, the proposed two-part evaluation scheme for economic feasibility should be applied to other WEEE streams and compared with results of earlier studies, which determined whether and in what way the recycling of certain WEEE streams is economically feasible. This could validate the usefulness of this evaluation scheme. Furthermore, the suggested evaluation dimensions—the upper crust concentrations and raw material prices—should be supplemented with additional dimensions and data to enhance the validity and expand the applicability of the scheme. Only data that are readily available and easy to collect should be integrated to preserve the main objective of the evaluation system: providing a fast and easy way to determine whether further investigations into the recyclability and feasibility are warranted and on which materials the focus should lie. Finally, this scheme could be applied to determine the economic feasibility of the recovery of materials other than elements. In this case, upper crust concentrations—which only relate to elements—could not be used. An alternative metric in addition to raw material prices would need to be applied.

## 5. Conclusions

The purpose of this study was to introduce a simple evaluation scheme that could be used to determine the potential and limitations of critical raw material recycling. The two-part evaluation system consisting of upper continental crust concentrations and raw material prices does not require much data collection effort. It represents a simple tool that can be applied to various WEEE streams and expanded to materials other than elements. The usefulness of the evaluation scheme was demonstrated in the case of LED lamps. In this context, this study also contributed to the LED literature. It addressed the research gap concerning the economic feasibility of LED lamp recycling, as mentioned by Cenci et al. [16] and Rahman et al. [10], with a focus on critical raw materials. Previous studies focused mainly on non-critical materials, disregarded collection and recycling rates when calculating the material mass available for recovery, and considered only raw material prices without comparing natural occurrence with material concentrations in LED lamp waste. Moreover, these investigations required high effort, a lot of time, and a lot of data input. These shortcomings were addressed in this study by examining the economic feasibility of recycling critical raw materials—specifically addressing the potential of indium, gallium, and rare earth elements—as well as accounting for losses during the collection and recycling steps.

The results of this study show that precious metals—particularly, gold—are the most economically viable materials contained in the LED part of an LED lamp. These materials are contained in higher concentrations in the lamp waste than the upper continental crust. They comprise high total masses, and they generate the most revenue out of the three different material groups investigated. Indium, gallium, and rare earth elements have low concentrations, low total masses, and generate low potential revenue. Therefore, new recycling technologies for LED lamps should focus on precious metals and be optimized to lose as little as possible of those elements in the process. Whether this amount of revenue would suffice to develop and implement an appropriate LED recycling technology needs to be investigated by a cost–benefit analysis considering the costs of the specific technology. The specific economic potential of the recycling of LED lamps depends on the recycling technology applied. Pre-treatment and pre-concentration steps that require manual labor will increase recycling costs. At the same time, not only recycling but also collection steps need to be considered. The currently low collection rate of 14% for waste lighting equipment in the EU shows that significant improvements are required to reach the EU collection target and to increase revenue through larger material quantities available for recycling.

Future research endeavors should include further studies on WEEE recycling, which leverage the herein proposed two-part evaluation scheme to validate its usefulness. More-

over, additional easily accessible metrics and data to estimate the economic feasibility of material recycling of other WEEE streams should be suggested.

**Author Contributions:** J.S.N. contributed to the conceptualization, methodology, validation, formal analysis, data curation, investigation, visualization, writing, review, and editing. M.R. added to the conceptualization, writing, review and editing, supervision, and project administration. N.v.G. made contributions to the conceptualization, methodology, investigation, review, and editing. All authors have read and agreed to the published version of the manuscript.

**Funding:** This research was funded by EIT RawMaterials as part of the project "REDLED: Recycling EnD-of-life LED" with the project number 18039. Financial support was provided by Wuppertal Institut für Klima, Umwelt, Energie gGmbH within the funding program Open Access Publishing.

**Institutional Review Board Statement:** Not applicable.

**Informed Consent Statement:** Not applicable.

**Data Availability Statement:** The data used to derive the results are freely available and contained in the article and Appendix A.

**Conflicts of Interest:** The authors declare no conflict of interest. The funders had no role in the design of the study; in the collection, analyses, or interpretation of data; in the writing of the manuscript, or in the decision to publish the results.

## Appendix A

The values in Table A1 between 2008 and 2020 relating to the total POM amounts as well as the different LED lamp applications were taken from Marwede et al. [9]. The values between 2021 and 2025 were extrapolated based on the data until 2020. Given the assumption by Buchert et al. [26] that the LED demand will reach 95% of the market share, the value in 2030 was calculated to represent 100%. The data for the years between 2025 and 2030 were interpolated.

**Table A1.** Yearly put-on-the-market (POM) amounts in units of lamps for different LED lamp applications in EU28 member states.

| Year | Residential (pcs) | Commercial (pcs) | Industrial (pcs) | Outdoor (pcs) | Architectural (pcs) | Residential Retrofits (pcs) | Commercial Retrofits (pcs) | Total POM (pcs) |
|---|---|---|---|---|---|---|---|---|
| 2030 | 576,816,079 | 73,412,955 | 10,487,565 | 10,487,565 | 20,975,130 | 314,626,952 | 41,950,260 | 1,048,756,507 |
| 2029 | 571,047,918 | 72,678,826 | 10,382,689 | 10,382,689 | 20,765,379 | 311,480,682 | 41,530,758 | 1,038,268,942 |
| 2028 | 565,279,757 | 71,944,696 | 10,277,814 | 10,277,814 | 20,555,628 | 308,334,413 | 41,111,255 | 1,027,781,377 |
| 2027 | 559,511,596 | 71,210,567 | 10,172,938 | 10,172,938 | 20,345,876 | 305,188,143 | 40,691,752 | 1,017,293,811 |
| 2026 | 553,743,436 | 70,476,437 | 10,068,062 | 10,068,062 | 20,136,125 | 302,041,874 | 40,272,250 | 1,006,806,246 |
| 2025 | 547,975,275 | 69,742,308 | 9,963,187 | 9,963,187 | 19,926,374 | 298,895,604 | 39,852,747 | 996,318,681 |
| 2024 | 513,482,418 | 65,352,308 | 9,336,044 | 9,336,044 | 18,672,088 | 280,081,319 | 37,344,176 | 933,604,396 |
| 2023 | 478,989,560 | 60,962,308 | 8,708,901 | 8,708,901 | 17,417,802 | 261,267,033 | 34,835,604 | 870,890,110 |
| 2022 | 444,496,703 | 56,572,308 | 8,081,758 | 8,081,758 | 16,163,516 | 242,452,747 | 32,327,033 | 808,175,824 |
| 2021 | 410,003,846 | 52,182,308 | 7,454,615 | 7,454,615 | 14,909,231 | 223,638,462 | 29,818,462 | 745,461,538 |
| 2020 | 379,500,000 | 48,300,000 | 6,900,000 | 6,900,000 | 13,800,000 | 207,000,000 | 27,600,000 | 690,000,000 |
| 2019 | 341,000,000 | 43,400,000 | 6,200,000 | 6,200,000 | 12,400,000 | 186,000,000 | 24,800,000 | 620,000,000 |
| 2018 | 319,000,000 | 40,600,000 | 5,800,000 | 5,800,000 | 11,600,000 | 174,000,000 | 23,200,000 | 580,000,000 |
| 2017 | 280,500,000 | 35,700,000 | 5,100,000 | 5,100,000 | 10,200,000 | 153,000,000 | 20,400,000 | 510,000,000 |
| 2016 | 247,500,000 | 31,500,000 | 4,500,000 | 4,500,000 | 9,000,000 | 135,000,000 | 18,000,000 | 450,000,000 |
| 2015 | 214,500,000 | 27,300,000 | 3,900,000 | 3,900,000 | 7,800,000 | 117,000,000 | 15,600,000 | 390,000,000 |
| 2014 | 159,500,000 | 20,300,000 | 2,900,000 | 2,900,000 | 5,800,000 | 87,000,000 | 11,600,000 | 290,000,000 |
| 2013 | 110,000,000 | 14,000,000 | 2,000,000 | 2,000,000 | 4,000,000 | 60,000,000 | 8,000,000 | 200,000,000 |
| 2012 | 60,500,000 | 7,700,000 | 1,100,000 | 1,100,000 | 2,200,000 | 33,000,000 | 4,400,000 | 110,000,000 |
| 2011 | 38,500,000 | 4,900,000 | 700,000 | 700,000 | 1,400,000 | 21,000,000 | 2,800,000 | 70,000,000 |
| 2010 | 24,200,000 | 3,080,000 | 440,000 | 440,000 | 880,000 | 13,200,000 | 1,760,000 | 44,000,000 |
| 2009 | 11,000,000 | 1,400,000 | 200,000 | 200,000 | 400,000 | 6,000,000 | 800,000 | 20,000,000 |
| 2008 | 5,500,000 | 700,000 | 100,000 | 100,000 | 200,000 | 3,000,000 | 400,000 | 10,000,000 |

The data for the development of the collection scenarios were taken from Eurostat and are displayed in Table A2. The collection rate is calculated according to the EU Directive

2012/19/EU [8] by dividing the waste collected in year t by the average POM amount of the three previous years. The data were accessed under https://ec.europa.eu/eurostat/databrowser/view/env_waselee/default/table?lang=en (accessed on 28 August 2019).

**Table A2.** Data used to calculate collection rates for lighting equipment in EU28 member states.

| Year | POM (t) | Waste Collected (t) | Collection Rate (%) |
|------|---------|---------------------|---------------------|
| 2017 | 518,852 | 68,940 | 14 |
| 2016 | 485,245 | 54,914 | 13 |
| 2015 | 560,470 | 36,713 | 10 |
| 2014 | 390,760 | 27,774 | 7 |
| 2013 | 350,599 | 24,955 | – |
| 2012 | 389,443 | 20,461 | – |
| 2011 | 379,300 | 18,185 | – |

**Table A3.** Yearly estimated collection and recycling rates for EU28 member states between 2017 and 2030.

| Year | Scenario 1 Rates (%) | | Scenario 2 Rates (%) | | Scenario 3 Rates (%) | |
|------|------------|-----------|------------|-----------|------------|-----------|
| | Collection | Recycling | Collection | Recycling | Collection | Recycling |
| 2030 | 14 | 50 | 50 | 65 | 85 | 80 |
| 2029 | 14 | 50 | 47 | 64 | 80 | 78 |
| 2028 | 14 | 50 | 45 | 63 | 74 | 75 |
| 2027 | 14 | 50 | 42 | 62 | 69 | 73 |
| 2026 | 14 | 50 | 39 | 60 | 63 | 71 |
| 2025 | 14 | 50 | 36 | 59 | 58 | 68 |
| 2024 | 14 | 50 | 34 | 58 | 52 | 66 |
| 2023 | 14 | 50 | 31 | 57 | 47 | 64 |
| 2022 | 14 | 50 | 28 | 56 | 42 | 62 |
| 2021 | 14 | 50 | 25 | 55 | 36 | 59 |
| 2020 | 14 | 50 | 23 | 53 | 31 | 57 |
| 2019 | 14 | 50 | 20 | 52 | 25 | 55 |
| 2018 | 14 | 50 | 17 | 51 | 20 | 52 |
| 2017 | 14 | 50 | 14 | 50 | 14 | 50 |

**Table A4.** Prices for critical raw materials from the Institute for Rare Earths and Strategic Metals for cerium, europium, gadolinium, and terbium [49–60], and from United States Geological Survey for gallium, gold, indium, palladium, silver, and yttrium [48].

| Month and Year | Cerium | Europium | Gadolinium | Gallium | Gold | Indium | Palladium | Silver | Terbium | Yttrium |
|----------------|--------|----------|------------|---------|------|--------|-----------|--------|---------|---------|
| | | | | | Prices in USD/kg | | | | | |
| October 2019 | 4.88 | N.A. [1] | 26.98 | – [2] | – | – | – | – | 716.82 | – |
| November 2019 | 4.97 | N.A. | 26.10 | – | – | – | – | – | 660.70 | – |
| December 2019 | 4.61 | N.A. | 26.98 | – | – | – | – | – | 628.28 | – |
| January 2020 | 4.66 | N.A. | 28.70 | – | – | – | – | – | 645.00 | – |
| February 2020 | 4.71 | N.A. | 28.58 | – | – | – | – | – | 645.81 | – |
| March 2020 | 4.76 | N.A. | 30.27 | – | – | – | – | – | 763.91 | – |
| April 2020 | 4.50 | 288.00 | N.A. | – | – | – | – | – | 715.00 | – |
| May 2020 | 4.50 | 288.00 | N.A. | – | – | – | – | – | 712.00 | – |
| June 2020 | 4.40 | 285.00 | N.A. | – | – | – | – | – | 820.00 | – |
| July 2020 | 4.35 | 285.00 | N.A. | – | – | – | – | – | 835.00 | – |

**Table A4.** *Cont.*

| Month and Year | Cerium | Europium | Gadolinium | Gallium | Gold | Indium | Palladium | Silver | Terbium | Yttrium |
|---|---|---|---|---|---|---|---|---|---|---|
| | Prices in USD/kg | | | | | | | | | |
| August 2020 | 4.35 | 286.00 | N.A. | – | – | – | – | – | 853.00 | – |
| September 2020 | 4.30 | 286.00 | N.A. | – | – | – | – | – | 920.00 | – |
| Yearly | 4.58 | 286.33 | 27.94 | 570.00 | 45,010.98 | 390.00 | 48,226.05 | 520.84 | 742.96 | 34.00 |

[1] N.A. signifies that the values for these dates were not available through the Institute for Rare Earths and Strategic Metals. [2] The dash (–) signifies values for these dates are not applicable to these elements because only yearly data were used from U.S.G.S.

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
