# Peer review of "The Potential and Limitations of Critical Raw Material Recycling: The Case of LED Lamps"

_resources, doi:10.3390/resources10040037_

Round 1

Reviewer 1 Report

Thank you for giving me the opportunity to review this manuscript. The work is of high quality and I recommend the publication as is. I made a few comments on the manuscript in order to increase the readability. Please see attached the manuscript. 

Author Response

The following changes were made to the manuscript:

  • In line 9 on p. 1, the wording was slightly changed of the sentence.
  • In line 40 on p. 1, the source in reference 4 was added at the end of the sentence and an additional citation of a reference was added in line 42 to provide the source document of the latest list of the EU defining the critical raw materials.
  • In line 115 on p. 3, the table heading was slightly adjusted from “material composition” to “material demand” to clarify that the contents of the table relate to the material requirements per 1 mm2 die area of white LED.
  • In line 121 on p. 3, the citation was changed from source 4 to source 6 to reference the latest list of the EU defining the critical raw materials. In line 233 on p. 6, the word “cumulative” was added to clarify what type of Weibull distribution function was used.
  • In lines 236 and 237, a sentence was added to clarify what this type of distribution function is usually used for.
  • In lines 322 to 324 on p. 9, a table footer was added for Table 6 to remind the reader of the definition of each scenario. A reference was made to the exact collection and recycling rates in the appendix as well.
  • In lines 372 to 374 on p. 10, a table footer was added to Table 8 to remind the reader of the time frame that the raw material prices refer to. A reference was also made to the appendix which contains more details on the raw material prices.
  • The sentences in lines 393 to 398 on pp. 10 and 11 were slightly adjusted to clarify that the recycling and recovery of rare earth elements and indium and gallium is not economically viable, unlike the recovery of precious metals which has a higher revenue potential.

The following comments by the reviewer were not addressed by changes in the manuscript:

  • We were unable to add an illustration of an LED lamp because we do not have a graphic or photograph ourselves and do not have any rights to such a document.
  • Regarding the question on the 100% material recovery values cited in Table 4: We cited the values from the referenced sources which investigated the recovery potential under laboratory conditions. We included ranges of values in our investigation to account for uncertainty in the recoveries under real world conditions. Adjustment of the values found in the literature is otherwise difficult because any percentage adjustment would be difficult to justify. Therefore, we have used what we found in the literature unchanged and added the results from various studies to account for uncertainty.
  • Regarding the Weibull distribution and gamma function: The gamma function is a special mathematical function whose exact definition is not mentioned in any of the sources we cited (and we researched additional papers which utilize the Weibull function and they do not define it either). It is widely used for calculations and modeling and even implemented as an easy-to-use function in Excel and other statistical software and programming languages. The specific definition in this section of this function does not add much insight, in our opinion, because this function is a basic mathematical concept. The definition is considered beyond the scope of this paper. If the reader is unfamiliar with the gamma function, a simple Google search may enlighten him or her more compared to the citation of a mathematical paper defining the foundations of this function.
  • Regarding the London Metal Exchange as a source for raw material prices: Using the LME data is obvious, but unfortunately, the LME only offers data for relatively few metals (at least publicly available). Especially for rare earths, no data is available there.
  • Regarding the question whether comparing upper crust concentrations can be compared to lamp waste concentrations: The metals are extracted either from primary or secondary sources. The basic processing and extraction steps from natural and anthropogenic deposits do not differ fundamentally. In both cases, after possible other processing and enrichment steps, hydro- or pyrometallurgical processes have to be applied. In this respect, a comparison based on the substance contents in natural and anthropogenic sources and deposits seems suitable to us for such a comparison.
  • Regarding the question on how rising raw material prices could affect economic viability: Even with significantly rising prices, it cannot be assumed that recycling will become economically viable, while extraction from primary raw materials will continue to be less costly.

Reviewer 2 Report

The manuscript is well written and the storyline is clear.

It is really a pity that LEDs are not properly recycled today and we are producing more an more waste. Therefore, there are already companies renting light, since they know their product best and can even repair lights, if needed. Next to the ecodesign approach I was missing the approach of the sharing economy.

I have only very few recommendations:

Table 6: Please give more information how you define the scenarios 1-3. I think this is needed in the paper and not in the ANNEX.

Table 8 and in the accompagnying text: Please give the reference year (or the time frame) of the USD price in the manuscript. You can then make a reference to the Annex.

In summary, the paper adresses an important topic today. It is a shame that we are producing so much waste. This does not fit to our Sustainable Development Goal Sustainable Consumption and the critical discussion with this issue could have been a bit more under the view of sustainable development and not only from the economic view.

Author Response

The following changes were made to the manuscript:

  • In line 9 on p. 1, the wording was slightly changed of the sentence.
  • In line 40 on p. 1, the source in reference 4 was added at the end of the sentence and an additional citation of a reference was added in line 42 to provide the source document of the latest list of the EU defining the critical raw materials.
  • In line 115 on p. 3, the table heading was slightly adjusted from “material composition” to “material demand” to clarify that the contents of the table relate to the material requirements per 1 mm2 die area of white LED.
  • In line 121 on p. 3, the citation was changed from source 4 to source 6 to reference the latest list of the EU defining the critical raw materials. In line 233 on p. 6, the word “cumulative” was added to clarify what type of Weibull distribution function was used.
  • In lines 236 and 237, a sentence was added to clarify what this type of distribution function is usually used for.
  • In lines 322 to 324 on p. 9, a table footer was added for Table 6 to remind the reader of the definition of each scenario. A reference was made to the exact collection and recycling rates in the appendix as well.
  • In lines 372 to 374 on p. 10, a table footer was added to Table 8 to remind the reader of the time frame that the raw material prices refer to. A reference was also made to the appendix which contains more details on the raw material prices.
  • The sentences in lines 393 to 398 on pp. 10 and 11 were slightly adjusted to clarify that the recycling and recovery of rare earth elements and indium and gallium is not economically viable, unlike the recovery of precious metals which has a higher revenue potential.

The following comments by the reviewer were not addressed by changes in the manuscript:

  • Regarding the comment on adding sharing economy into the discussion section: There are further alternative approaches which could have a stronger environmental and economic benefit than recycling of LED lamps. However, we only wanted to provide some brief context and leave closer examination of alternatives to future research papers.
  • Regarding the comment on integrating a view of sustainable development in addition to the economic view: The view of sustainable development is necessary for a comprehensive evaluation of whether certain actions, incl. recycling, are desirable and should be incentivized by governmental actors. However, our evaluation specifically wanted to highlight the economic viability, not the environmental or social perspective, to have a targeted research question and to fill the gap in the literature which was lacking a systemic approach to assess economic viability. We think future research endeavors can build on our findings and integrate sustainable development considerations.
